# Comparison of Endurance Time Prediction of Biceps Brachii Using Logarithmic Parameters of a Surface Electromyogram during Low-Moderate Level Isotonic Contractions

**Chang-ok Cho** [1] **, Jin-Hyoung Jeong** [2] **, Yun-jeong Kim** [3] **, Jee Hun Jang** [3] **, Sang-Sik Lee** [4,*] **and Ki-young Lee** [4,*]

[1]   Korea Paralympic Committee, Seoul 05540, Korea; cco7171@hanmail.net
[2]   Department of Biomedical IT, Catholic Kwandong University, Gangneung-si 25601, Korea; jjh830813@naver.com
[3]   Department of Sport and Leisure Studies, Catholic Kwandong University, Gangneung-si 25601, Korea; yunjeong45@naver.com (Y.-j.K.); jjh@cku.ac.kr (J.H.J.)
[4]   Department of Biomedical Engineering, Catholic Kwandong University, Gangneung-si 25601, Korea
*   Correspondence: lsskyj@cku.ac.kr (S.-S.L.); kylee@cku.ac.kr (K.-y.L.)

**Abstract:** At relatively low effort level tasks, surface electromyogram (sEMG) spectral parameters have demonstrated an inconsistent ability to monitor localized muscle fatigue and predict endurance capacity. The main purpose of this study was to assess the potential of the endurance time ($T_{end}$) prediction using logarithmic parameters compared to raw data. Ten healthy subjects performed five sets of voluntary isotonic contractions until their exhaustion at 20% of their maximum voluntary contraction (MVC) level. We extracted five sEMG spectral parameters namely the power in the low frequency band (LFB), the mean power frequency (MPF), the high-to-low ratio between two frequency bands (H/L-FB), the Dimitrov spectral index (DSI), and the high-to-low ratio between two spectral moments (H/L-SM), and then converted them to logarithms. Changes in these ten parameters were monitored using area ratio and linear regressive slope as statistical predictors and estimating from onset at every 10% of $T_{end}$. Significant correlations ($r > 0.5$) were found between $\log(T_{end})$ and the linear regressive slopes in the logarithmic H/L-SM at every 10% of $T_{end}$. In conclusion, logarithmic parameters can be used to describe changes in the fatigue content of sEMG and can be employed as a better predictor of $T_{end}$ in comparison to the raw parameters.

**Keywords:** electromyography; muscle; endurance capacity; isotonic; prediction capability

## 1. Introduction

In everyday life, low-moderate level isotonic exercise is the natural way of human activity and includes a concentric contraction and an eccentric contraction. Concentric contractions are the primary functions of biceps brachii muscles, and endurance contractions primarily work to slow twitch fibers and develop such fibers in their efficiency and resistance to fatigue [1]. Fatigue can be defined as the exercise-induced decrease in the ability to produce force [2] and has been measured by using surface electromyography (sEMG) as an assessment tool in prevention, monitoring, and rehabilitation fields [3].

Endurance capacity is the ability to sustain a given force over time, while measurement of the endurance time ($T_{end}$) is an indicator of the muscle resistance to fatigue [4–6]. Although widely used in clinical practice, it is problematic to measure the effect of physical and psychological factors such as pain and motivation [7,8]. Thus, methods that enable reliable estimates of muscle endurance time during the time shorter than the endurance time are of great importance for studying muscle function and motor control. A lot of researchers have studied endurance time prediction due to the fact that firing statistics of the active motor units (MU) were shown to affect the sEMG power spectrum toward lower frequencies as spectral compression [9–12]. In addition, sEMG has been shown to be a more objective approach to measuring muscle fatigue which is generally accompanied by an

increase in amplitude of the sEMG signal because of the firing rates of increased motor unit recruitment [13,14]. Badier et al. (1993) found a significant relationship between $T_{end}$ and the time-constant of a high-to-low ratio with the fixed frequency band as computed within the first 10–20 s of contraction [15]. Hanayama (1994) found no significant correlation beween $T_{end}$ and the decreasing changes of muscle fiber conduction velocity (MFCV) [16]. After that, extrapolation of $T_{end}$ based on linear regressive slopes of sEMG power spectrum has been reliable when computed over submaximal durations more than 50% $T_{end}$ whatever the level of contraction considered [17–20]. In addition, Maïsetti et al. (2002) demonstrated that the area ratio which was proposed by Merletti et al. (1991) as changes of the low frequency band (LFB), as estimated around the first 25% of $T_{end}$, were significantly correlated with $T_{end}$. Lee et al. (2011) found the sustained times around 31% of $T_{end}$, when the Dimitrov spectral index (DSI) (Dimitrov et al. 2006) was above 130% of the first value, were significantly correlated with $T_{end}$. Lee et al. (2017) proposed high-to-low ratio between two signal spectral moments without choosing the optimal border frequencies of the low and high bands (H/L-SM). The experimental result obtained showed that linear regressive slopes of H/L-SM over the first 30% of $T_{end}$ were significantly correlated with $T_{end}$ [21–23].

Mean power frequency (MPF), median frequency (MDF), and high-to-low ratio with fixed frequency band (H/L-FB) are proposed as the spectral parameters related to the spectral compression of the sEMG signal, which decline throughout fatigue trials [24–26]. However, their consistent changes have been documented especially for relative high effort levels [27]. In contrast, these parameters have yielded an inconsistent pattern during sustained contractions at low level efforts. González-Izal et al. (2010) employed the logarithmic transformation of DSI as a predictor of the performance change in muscle power to reduce the large variability [28]. Lee et al. (2017; 2019) proposed the H/L-SM and converted it to logarithms to monitor the more sensitive activity of biceps femoris muscles during treadmill walking [23,29]. Yassierli and Nussbaum (2003; 2008) demonstrated that the Poisson-fit model using the logarithmic transformation could be more sensitive in localized muscle fatigue in sEMG-based assessments [30,31]. In myoelectric pattern recognition, logarithmic parameters in sEMG are especially useful to decode limb movements regarding the control of powered prostheses [32]. To our knowledge, the ability to predict $T_{end}$ using logarithmic parameters at submaximal time periods shorter than $T_{end}$ during the isotonic contraction test is limited.

Our study was designed to test whether changes in the logarithmic parameters calculated over a shorter duration than $T_{end}$ could predict the endurance time of the biceps brachii muscle. Thus, sEMG parameters such as LFB, MPF, H/L-FB, DSI, and H/L-SM were converted to logarithms, and two types of changes were calculated by using the area ratio and the slope of linear regression model as predictors of $T_{end}$. Subsequently, the relationships between $T_{end}$ and predictors were analyzed and evaluated.

## 2. Materials and Methods

### 2.1. Subjects

Ten healthy subjects (5 males and 5 females) with no history of cardiovascular, neurological, and musculoskeletal disorders, volunteered for this study. Their demographics (age, height, and mass) were measured and are described in Table 1. The subjects were informed of the purpose of the study before their consent was obtained. This study was approved by the Institutional Bioethics Committee of the Catholic Kwandong University, South Korea.

**Table 1.** Subject demographics data.

| Variable | Mean | Standard Deviation |
|:---:|:---:|:---:|
| Age (yrs) | 26.0 | 2.7 |
| Height (cm) | 165.4 | 6.2 |
| Weight (kg) | 63.7 | 12.5 |

## 2.2. Apparatus

### 2.2.1. MMT

The manual muscle tester (MMT) (Model: 01163, Manufacture: Lafayette Instrument Company, Sagamore Pkwy, IN, USA) was used to measure the maximal voluntary contraction (MVC) in accordance with the manufacture's manual (Figure 1).

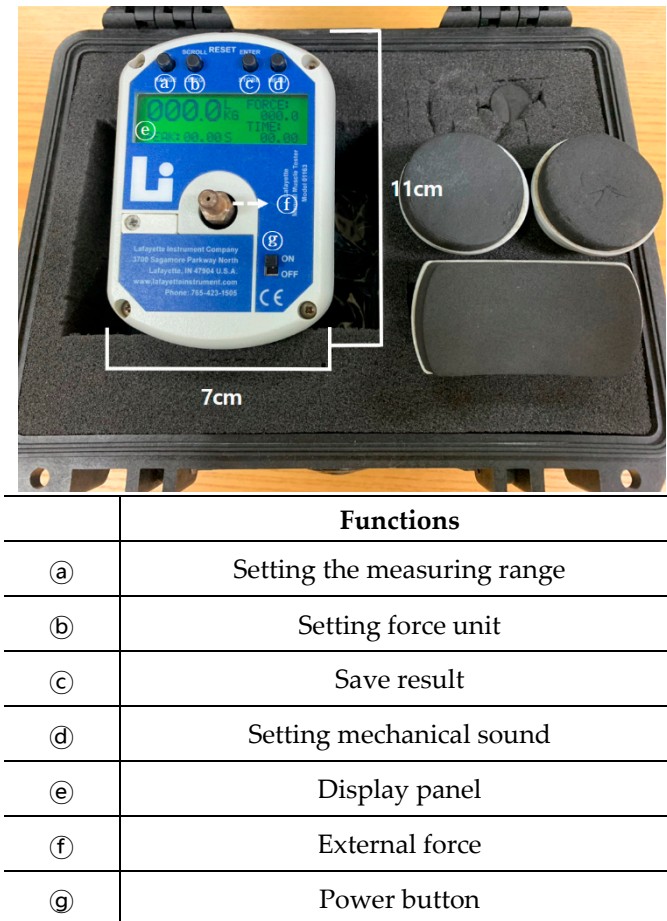

| | Functions |
|---|---|
| ⓐ | Setting the measuring range |
| ⓑ | Setting force unit |
| ⓒ | Save result |
| ⓓ | Setting mechanical sound |
| ⓔ | Display panel |
| ⓕ | External force |
| ⓖ | Power button |

**Figure 1.** Manual muscle tester.

The arm was at 110° flexion under the forearm in neutral position to measure the MVC of the subject using the MMT. The subject performed three maximal contractions 3 s long with 3 min rest period between them. The MVC was determined as the highest measured value.

### 2.2.2. Electromyography (EMG)

Surface EMG recordings were obtained from the biceps brachii muscles using bipolar surface electrodes (2 cm apart), which were connected to the measuring apparatus MyoTrace 400 with MyoResearch 3.6 software (Noraxon, AZ, USA). The sampling frequency was set at 1 kHz and boundaries of the band pass filter were set at 6 and 500 Hz (Figure 2). The electrodes were placed on the skin with anti-allergic tape after the skin was cleaned with alcohol and placed on the area of greatest muscle bulk along the longitudinal midline of the muscle [33,34].

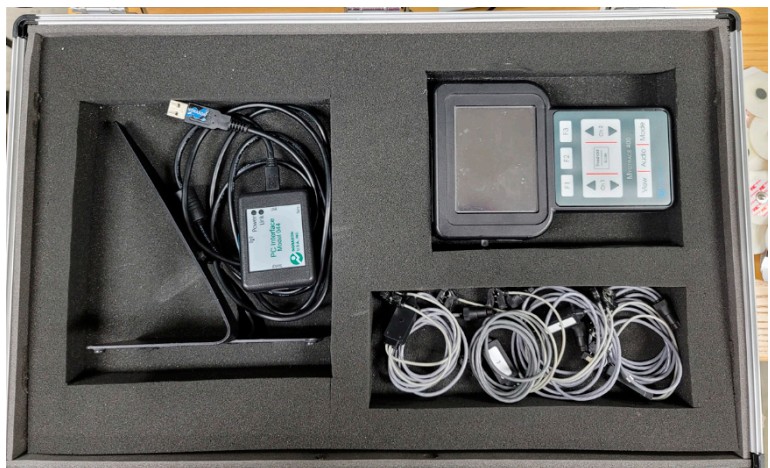

**Figure 2.** Surface EMG 4-channel wired instrument (MyoTrace 400).

*2.3. Experimental Protocol*

A schematic diagram of isotonic contraction of the biceps brachii muscles is shown in Figure 3. The subjects were asked to stand erect with their upper arm fixed and to move their lower arm through a range of motion from full extension to 110° flexion at a speed of 25 repetitions per minute using a metronome. Each repeated contraction was observed by an investigator and was considered successful if performed with the full range of motion within the metronome-guided time interval (2.4 s). During one set of the isotonic contraction trials, the subject was asked to continue repetitive contractions until exhaustion. The time of termination was determined when the participant indicated that they could no longer continue the full range of motion with the metronome speed for more than two repetitions, despite verbal encouragement without threats. This time point was noted as the $T_{end}$ for each subject. Ten subjects completed five sets of the isotonic contraction trials until their exhaustion at 20% MVC. Two hours of rest was provided between three sets of trials conducted over 1 day, and the subsequent two sets of trials were conducted after 3 days to avoid fatigue.

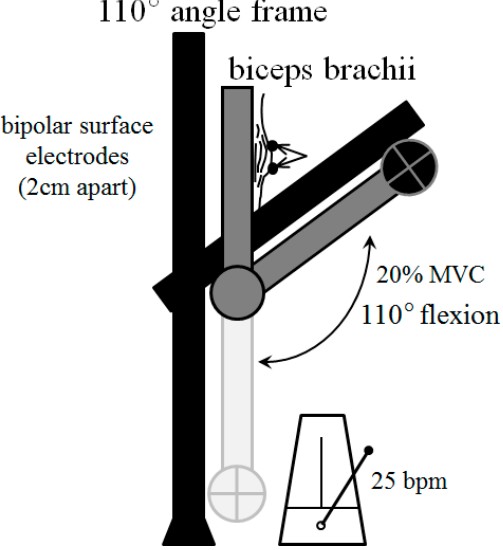

**Figure 3.** A schematic diagram of the isotonic contraction of the biceps brachii muscles.

## 2.4. Surface EMG Signal Acquisition

Surface EMG signals were collected using MyoResearch 3.6 software which guided the data acquisition steps. Figures 4 and 5 show the initial screen using this software, and the time-based graphic screen in the real-time progress, respectively.

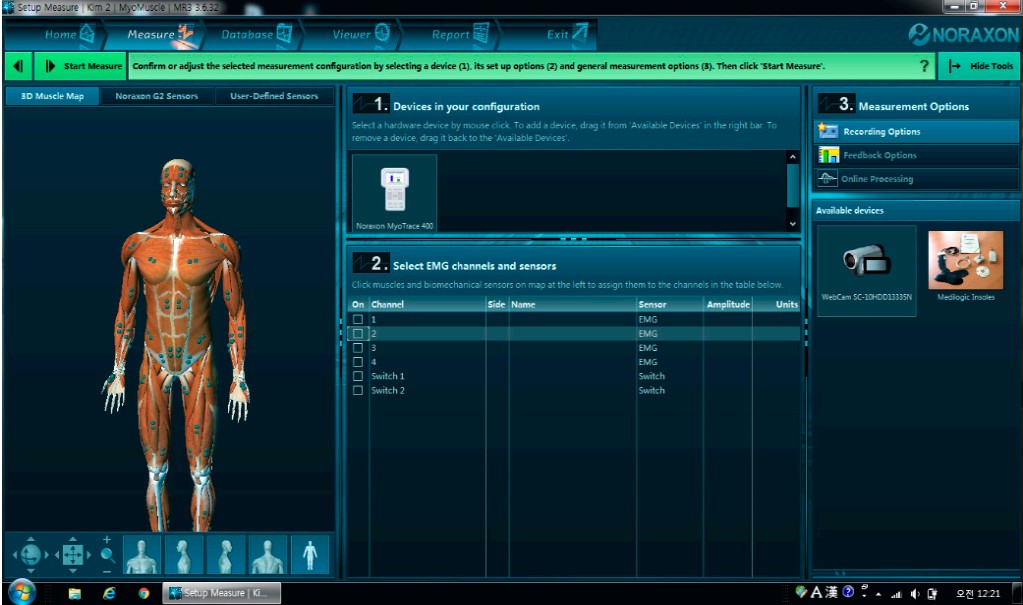

**Figure 4.** The initial screen.

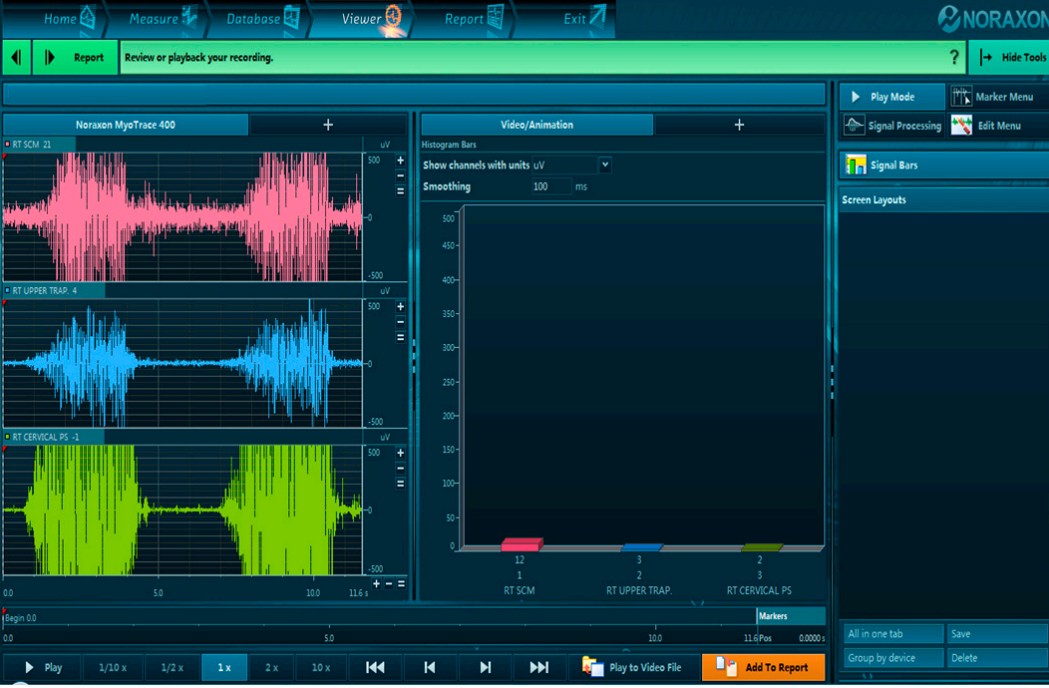

**Figure 5.** Surface EMG signal screen.

### 2.5. Mathematical Models

In this study, we used the five raw parameters from the sEMG power spectrum namely LFB, MPF, H/L-FB, DSI, and H/L-SM, and then converted these parameters to logarithms. The definitions of these logarithmic parameters are as follows [35].

$$\text{logLFB} = \log\left\{\sum_{f=f_{L1}}^{f_{L2}} P(f)\right\} \tag{1}$$

$$\text{logMPF} = \log\left\{\frac{\sum_{f=f_0}^{f_s/2} fP(f)}{\sum_{f=f_0}^{f_s/2} P(f)}\right\} \tag{2}$$

$$\text{logH/L} - \text{FB} = \log\left\{\frac{\sum_{f=f_{H1}}^{f_{H2}} P(f)}{\sum_{f=f_{L1}}^{f_{L2}} P(f)}\right\} \tag{3}$$

$$\text{logDSI} = \log\left\{\frac{\sum_{f=f_0}^{f_s/2} f^{-1}P(f)}{\sum_{f=f_0}^{f_s/2} f^5 P(f)}\right\} \tag{4}$$

$$\text{logH/L} - \text{SM} = \log\left\{\frac{\sum_{f=f_0}^{f_s/2} f^5 P(f)}{\sum_{f=f_0}^{f_s/2} f^{-1}P(f)}\right\} \tag{5}$$

Here,

$f_{L1} = 15$ Hz, $f_{L2} = 45$ Hz and $(f)$: power spectrum in expression (1);
$f_0 = 6$ Hz and $\frac{f_s}{2} = 500$ Hz in expression (2), (4) and (5);
$f_{H1} = 95$ Hz, $f_{H2} = 500$ Hz, $f_{L1} = 15$ Hz, $f_{L2} = 45$ Hz in expression (3).

In the curly brackets of Expressions (1)–(5), first, LFB in expression (1) is the power in the low frequency band of the sEMG power spectrum. MPF in expression (2) refers to the high-to-low ratio between the order 1 and the order 0 spectral moments as a measure of the change in muscle fiber propagation velocity. H/L-FB in expression (3) is the high-to-low ratio between two high and low bands with fixed border frequencies, whereas, DSI in expression (4) is the order ($-1$) spectral moment normalized by the order 5 moment, while DSI revealed a more notable change in muscle fatigue than MPF. Lastly, in expression (5), H/L-SM is similar to H/L-FB, and could be calculated without the fixed border frequencies. Following the definitions in expressions (1)–(5), these five sEMG spectral parameters were converted to logarithms. Logarithmic transformation has been widely used in biomedical and psychosocial research to deal with inconsistent data [36].

### 2.6. Data Analysis

Data analysis was performed using personal computer. For accurate spectral analysis of the sEMG signals (Figure 6) in isotonic contraction cycles (2.4 s), we used a 1 s time Hamming window every 0.3 s. Short-time Fourier transformation was conducted on each windowed segment to calculate the power spectrum, which was used to estimate the raw parameters such as LFB, MPF, H/L-FB, DSI, and H/L-SM in the curly brackets of expression (1)–(5). These parameters except MPF were normalized and expressed as percentages of initial values, and converted to logarithms. The coefficient of variation (CV) is known as the relative standard deviation and defined as the ratio of the standard deviation to the mean [37]. We used the CV to compare variability between the five raw and the five logarithmic parameters.

The $T_{end}$ of each subject was divided into 10 equal intervals at every 10% of $T_{end}$ to evaluate the relationships between $T_{end}$ and the statistical predictors such as the area ratio and the slope of linear regression model as estimated over the shorter periods than the $T_{end}$ [38]. For this purpose, we used the predictors as follows.

1. The area ratios in the five raw parameters
2. The area ratios in the five logarithmic parameters
3. The slopes in the five raw parameters
4. The slopes in the five logarithmic parameters

These four predictors were estimated over the periods from the onset to every 10% of $T_{end}$. The one-way ANOVA was used to compare the changes of each predictor according to the 10 periods. Pearson's correlation coefficient was used to quantify the performance of the relationships between $T_{end}$ and these values. The level of significance was set at $p < 0.05$.

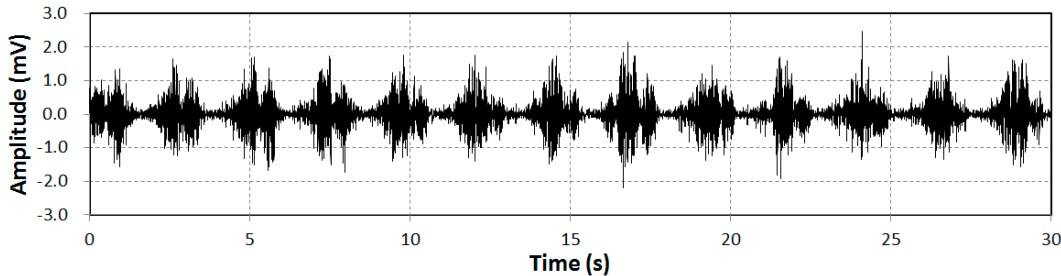

**Figure 6.** An example of the sEMG signal during isotonic contraction.

## 3. Results

### 3.1. MVC and $T_{end}$

MVC was 19.1 (5.9) kgf and $T_{end}$ was 53.7 (19.6) s during isotonic contractions at 20% MVC. Endurance times were sorted in descending order and displayed according to the isotonic contraction sets of the 10 subjects in Figure 7.

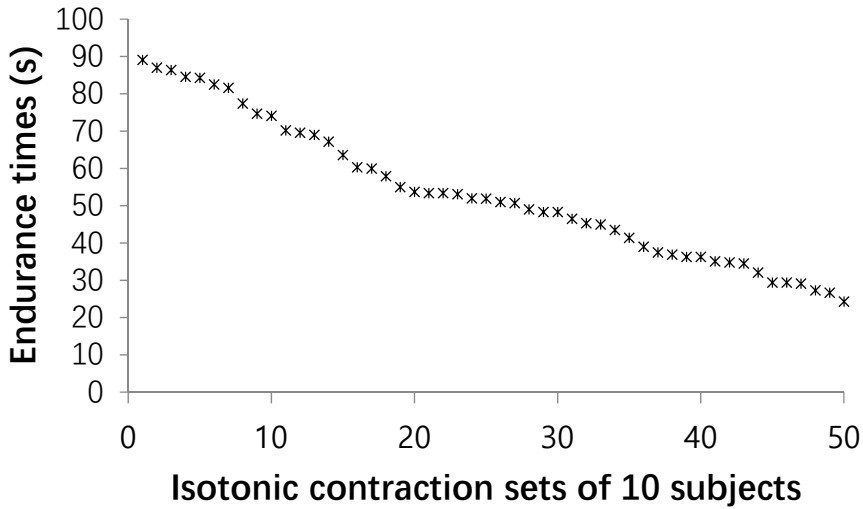

**Figure 7.** Endurance times in descending order.

### 3.2. Changes in sEMG Parameters and Predictors

Figure 8 showed the five raw and the five logarithmic parameters over the whole endurance time ($T_{end}$) for the subject whose $T_{end}$ was almost the same as the mean of the endurance times of all subjects. The left column (a) displays the five raw parameters, and the right column (b) displays the five logarithmic ones with respect to time. The ripple period in LFB and logLFB time series in Figure 8a,b is 2.4 s which can be calculated as 50 s divided by 21 ripples, and the same as the repeated period during the isotonic contractions.

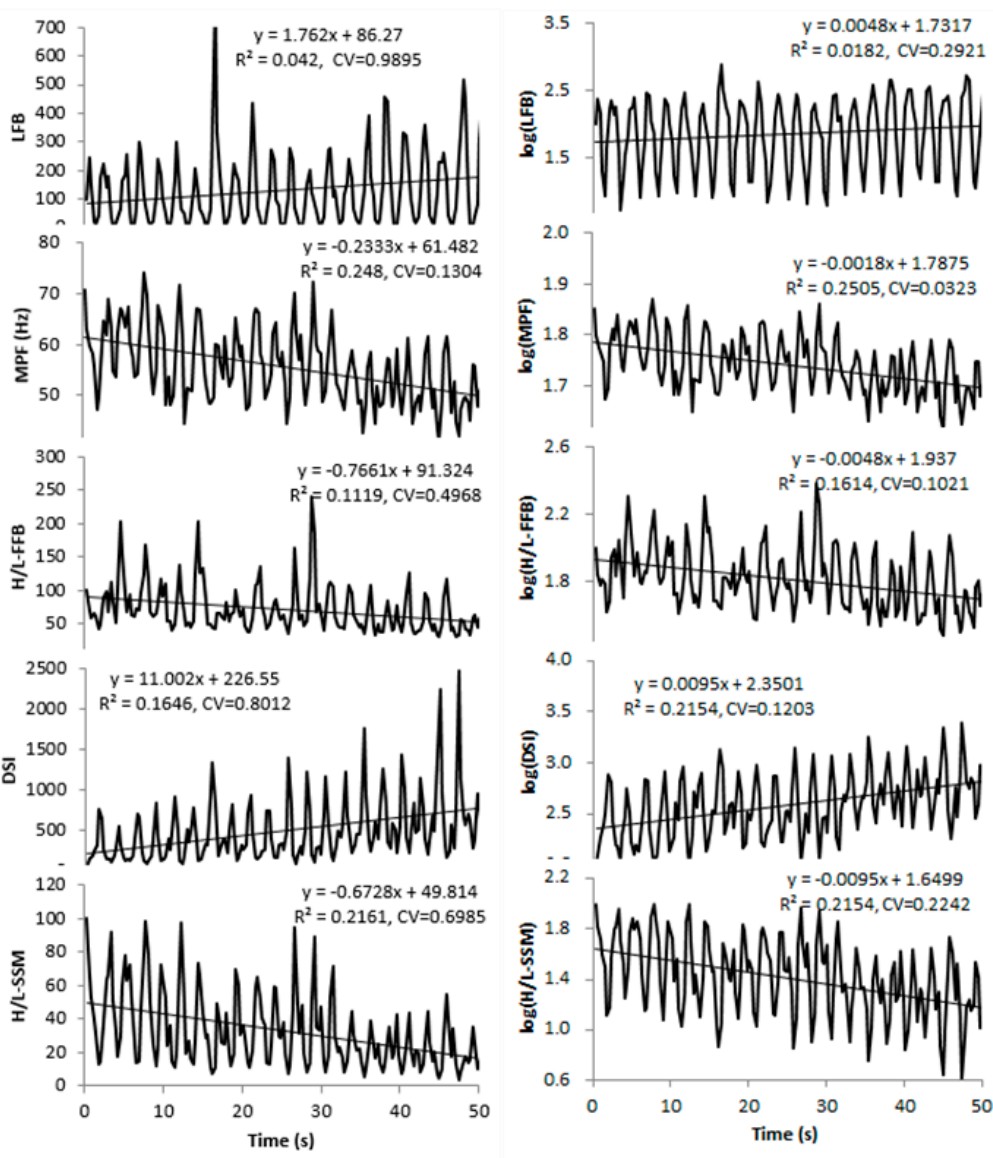

(**a**)The five raw parameters (**b**) The five logarithmic parameters

**Figure 8.** The time series of sEMG parameters during the endurance time ($T_{end}$) for the subject whose $T_{end}$ was almost the same as the mean for all measured sEMG of all subjects. The left column (**a**): the five raw parameters; the right column (**b**): the five logarithmic parameters; R2: coefficient of determination; CV: coefficient of variation.

The time series of LFB, DSI, logLFB, and logDSI increased, and those of the other parameters decreased during the endurance contractions, because muscle fatigue is generally accompanied by an increased firing rate of motor unit recruitment and spectral parameters related to spectral compression during static and dynamic contractions [39]. Similar results were reported in previous studies [9,23,27].

The CVs of the five raw and the five logarithmic parameters are compared in Table 2 which shows that the CV of the logarithmic parameter is less than that of the raw one. These results revealed that the logarithmic transformation could reduce the large variability in the raw parameter.

Figures 9 and 10 with Tables 3 and 4 show the time series of two predictors namely the area ratio and the slope in the raw and the logarithmic parameters as estimated over every period of 10% of $T_{end}$, respectively. In Figure 9 and Table 3, the area ratios in LFB, DSI, logLFB, and logDSI decreased linearly, and those in the others increased linearly, because

the definition varied between 0 and 1 for decreasing patterns and is negative for increasing patterns. In contrast, slopes in LFB, DSI, logLFB, and logD SI decayed exponentially, and those in the others rose exponentially as shown in Figure 10 and Table 4. One-way ANOVA was conducted on changes with respect to every period of 10% $T_{end}$ in each parameter. There were significant differences for the two predictors of all parameters ($p < 0.05$). Thus, these results showed that the time series of the area ratios and the slopes of the raw and the logarithmic parameters varied independently.

**Table 2.** Mean and standard deviation of coefficient of variation (CV) of all parameters.

| - | LFB | MPF | H/L-FB | DSI | H/L-SM |
|---|---|---|---|---|---|
| Raw | $0.27 \pm 0.51$ | $0.19 \pm 0.11$ | $0.20 \pm 0.15$ | $0.26 \pm 0.24$ | $0.20 \pm 0.18$ |
| Logarithm | $0.12 \pm 0.04$ | $0.17 \pm 0.13$ | $0.12 \pm 0.05$ | $0.14 \pm 0.08$ | $0.17 \pm 0.09$ |

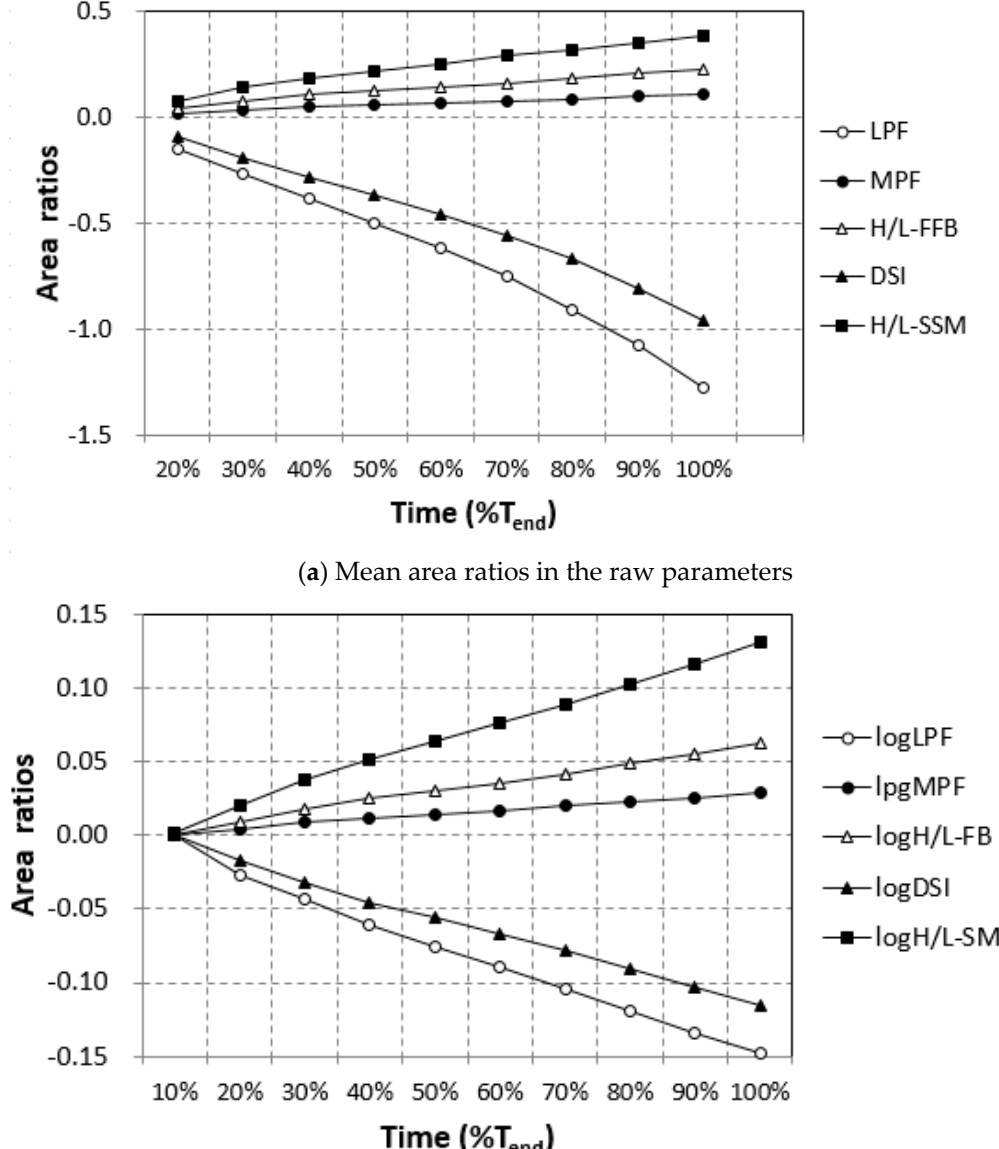

(**a**) Mean area ratios in the raw parameters

(**b**) Mean area ratios in the logarithmic parameters

**Figure 9.** The time series of area ratios in the sEMG raw parameters (**a**) and logarithmic ones (**b**) with respect to time as estimated over every 10% of $T_{end}$ (White circle: LFB and logLFB; Black circle: MPF and logMPF; White triangle: H/L-FB and logH/L-FB; Black triangle: DSI and logDSI; Black square: H/L-SM and logH/L-SM).

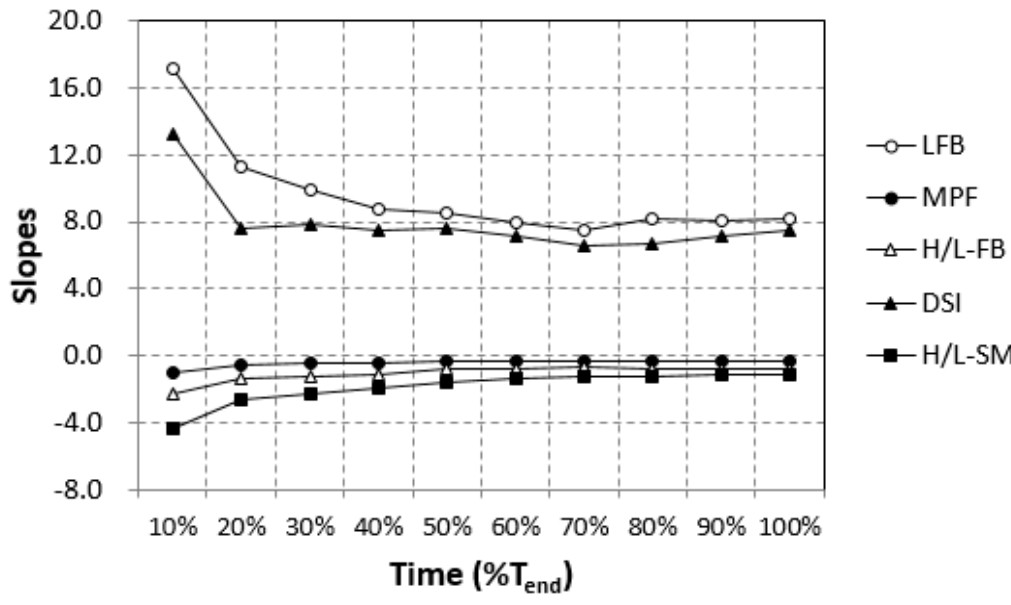

(**a**) Mean slopes in the raw parameters

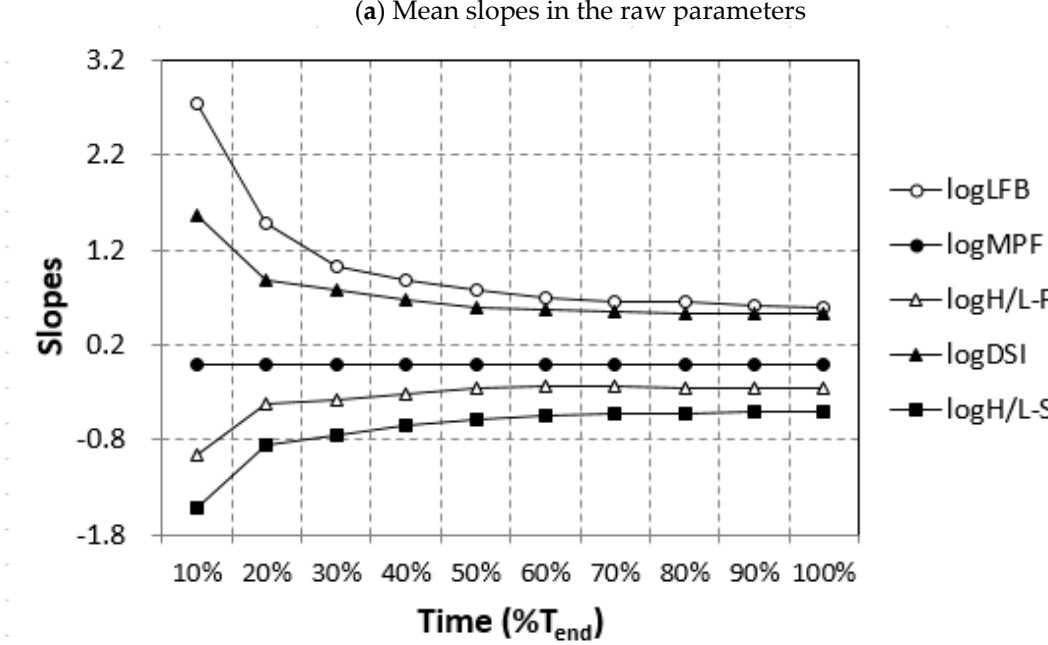

(**b**) Mean slopes in the logarithmic parameters

**Figure 10.** The time series of slopes in the sEMG raw parameters (**a**) and logarithmic ones (**b**) with respect to time as estimated over every 10% of $T_{end}$ (White circle: LFB and logLFB; Black circle: MPF and logMPF; White triangle: H/L-FB and logH/L-FB; Black triangle: DSI and logDSI; Black square: H/L-SM and logH/L-SM).

**Table 3.** Area ratios (mean and S.D.) in the sEMG raw parameters (a) and logarithmic ones (b) with respect to time as estimated over every 10% of $T_{end}$ (S.D.: standard deviation).

| | (a) Mean area ratios in the raw parameters | | | | | | | | | |
|---|---|---|---|---|---|---|---|---|---|---|
| | **%Tend** | **10%** | **20%** | **30%** | **40%** | **50%** | **60%** | **70%** | **80%** | **90%** | **100%** |
| LFB | Mean | 0.00 | −0.15 | −0.27 | −0.39 | −0.50 | −0.62 | −0.75 | −0.91 | −1.07 | −1.27 |
| | S.D. | 0.00 | 0.17 | 0.23 | 0.26 | 0.31 | 0.36 | 0.41 | 0.48 | 0.57 | 0.71 |
| MPF | Mean | 0.00 | 0.02 | 0.03 | 0.05 | 0.05 | 0.06 | 0.08 | 0.09 | 0.10 | 0.11 |
| | S.D. | 0.00 | 0.02 | 0.03 | 0.03 | 0.04 | 0.04 | 0.04 | 0.04 | 0.04 | 0.04 |
| H/L-FB | Mean | 0.00 | 0.04 | 0.08 | 0.10 | 0.12 | 0.14 | 0.16 | 0.18 | 0.21 | 0.23 |
| | S.D. | 0.00 | 0.07 | 0.09 | 0.09 | 0.10 | 0.10 | 0.11 | 0.11 | 0.11 | 0.10 |
| DSI | Mean | 0.00 | −0.09 | −0.19 | −0.28 | −0.37 | −0.46 | −0.56 | −0.67 | −0.81 | −0.96 |
| | S.D. | 0.00 | 0.10 | 0.16 | 0.24 | 0.33 | 0.34 | 0.36 | 0.38 | 0.46 | 0.49 |
| H/L-SM | Mean | 0.00 | 0.07 | 0.14 | 0.18 | 0.22 | 0.25 | 0.29 | 0.32 | 0.35 | 0.38 |
| | S.D. | 0.00 | 0.06 | 0.08 | 0.10 | 0.10 | 0.11 | 0.11 | 0.12 | 0.11 | 0.11 |
| | (b) Mean area ratios in the logarithmic parameters | | | | | | | | | |
| | **%Tend** | **10%** | **20%** | **30%** | **40%** | **50%** | **60%** | **70%** | **80%** | **90%** | **100%** |
| logLFB | Mean | 0.00 | −0.03 | −0.04 | −0.06 | −0.08 | −0.09 | −0.10 | −0.12 | −0.13 | −0.15 |
| | S.D. | 0.00 | 0.03 | 0.04 | 0.04 | 0.05 | 0.05 | 0.05 | 0.05 | 0.05 | 0.06 |
| logMPF | Mean | 0.000 | 0.004 | 0.009 | 0.012 | 0.014 | 0.017 | 0.020 | 0.023 | 0.026 | 0.029 |
| | S.D. | 0.000 | 0.006 | 0.008 | 0.009 | 0.011 | 0.011 | 0.011 | 0.011 | 0.012 | 0.012 |
| logH/L-FB | Mean | 0.00 | 0.01 | 0.02 | 0.02 | 0.03 | 0.03 | 0.04 | 0.05 | 0.06 | 0.06 |
| | S.D. | 0.00 | 0.02 | 0.02 | 0.02 | 0.02 | 0.03 | 0.03 | 0.03 | 0.03 | 0.03 |
| logDSI | Mean | 0.00 | −0.02 | −0.03 | −0.05 | −0.06 | −0.07 | −0.08 | −0.09 | −0.10 | −0.12 |
| | S.D. | 0.00 | 0.01 | 0.02 | 0.03 | 0.03 | 0.03 | 0.04 | 0.04 | 0.04 | 0.04 |
| logH/L-SM | Mean | 0.00 | 0.02 | 0.04 | 0.05 | 0.06 | 0.08 | 0.09 | 0.10 | 0.12 | 0.13 |
| | S.D. | 0.00 | 0.02 | 0.02 | 0.03 | 0.04 | 0.04 | 0.04 | 0.04 | 0.04 | 0.05 |

**Table 4.** Slopes (mean and S.D.) in the sEMG raw parameters (a) and logarithmic ones (b) with respect to time as estimated over every 10% of $T_{end}$ (S.D.: standard deviation).

| | (a) Mean slopes in the raw parameters | | | | | | | | | |
|---|---|---|---|---|---|---|---|---|---|---|---|
| | **%Tend** | **10** | **20** | **30** | **40** | **50** | **60** | **70** | **80** | **90** | **100** |
| LFB | Mean | 17.09 | 11.25 | 9.90 | 8.79 | 8.48 | 7.99 | 7.54 | 8.18 | 8.10 | 8.21 |
| | S.D. | 28.55 | 14.74 | 13.83 | 13.15 | 11.72 | 11.01 | 8.70 | 9.46 | 8.90 | 8.02 |
| MPF | Mean | −1.02 | −0.55 | −0.49 | −0.39 | −0.32 | −0.30 | −0.29 | −0.28 | −0.28 | −0.27 |
| | S.D. | 1.82 | 0.63 | 0.39 | 0.31 | 0.27 | 0.22 | 0.18 | 0.16 | 0.15 | 0.13 |
| H/L-FB | Mean | −2.27 | −1.35 | −1.25 | −1.11 | −0.83 | −0.73 | −0.72 | −0.76 | −0.76 | −0.76 |
| | S.D. | 8.36 | 2.48 | 1.20 | 0.78 | 0.67 | 0.58 | 0.52 | 0.49 | 0.46 | 0.40 |
| DSI | Mean | 13.24 | 7.65 | 7.82 | 7.51 | 7.62 | 7.09 | 6.55 | 6.67 | 7.17 | 7.45 |
| | S.D. | 20.77 | 8.53 | 10.01 | 11.32 | 12.19 | 7.77 | 5.28 | 4.89 | 5.80 | 5.21 |
| H/L-SM | Mean | −4.30 | −2.64 | −2.33 | −1.93 | −1.53 | −1.39 | −1.30 | −1.23 | −1.17 | −1.12 |
| | S.D. | 8.63 | 2.42 | 1.44 | 1.29 | 0.84 | 0.78 | 0.71 | 0.65 | 0.61 | 0.58 |
| | (b) Mean slopes in the logarithmic parameters | | | | | | | | | |
| | **%Tend** | **10%** | **20%** | **30%** | **40%** | **50%** | **60%** | **70%** | **80%** | **90%** | **100%** |
| logLFB | Mean | 2.74 | 1.48 | 1.04 | 0.88 | 0.77 | 0.69 | 0.66 | 0.65 | 0.61 | 0.59 |
| | S.D. | 5.70 | 2.01 | 1.10 | 0.74 | 0.69 | 0.52 | 0.44 | 0.37 | 0.30 | 0.25 |
| logMPF | Mean | −0.007 | −0.004 | −0.004 | −0.003 | −0.002 | −0.002 | −0.002 | −0.002 | −0.002 | −0.002 |
| | S.D. | 0.013 | 0.005 | 0.003 | 0.003 | 0.002 | 0.002 | 0.001 | 0.001 | 0.001 | 0.001 |
| logH/L-FB | Mean | −0.95 | −0.43 | −0.38 | −0.32 | −0.26 | −0.23 | −0.24 | −0.25 | −0.26 | −0.26 |
| | S.D. | 2.59 | 0.79 | 0.39 | 0.25 | 0.20 | 0.18 | 0.17 | 0.17 | 0.16 | 0.14 |

**Table 4.** *Cont.*

| | | | | | | | | | | | |
|---|---|---|---|---|---|---|---|---|---|---|---|
| logDSI | Mean | 1.57 | 0.88 | 0.78 | 0.68 | 0.60 | 0.57 | 0.54 | 0.54 | 0.53 | 0.53 |
| | S.D. | 2.39 | 0.77 | 0.54 | 0.49 | 0.44 | 0.39 | 0.34 | 0.32 | 0.30 | 0.26 |
| logH/L-SM | Mean | −1.52 | −0.85 | −0.75 | −0.66 | −0.58 | −0.54 | −0.52 | −0.52 | −0.51 | −0.51 |
| | S.D. | 2.44 | 0.77 | 0.55 | 0.49 | 0.42 | 0.36 | 0.30 | 0.27 | 0.25 | 0.21 |

### 3.3. Relationships between $T_{end}$ and Predictors

The correlation coefficients between $T_{end}$ and the changes in the raw and the logarithmic parameters as computed over every 10% of $T_{end}$ are shown in Tables 5 and 6. When the changes were estimated using the area ratio, no significant correlations between the endurance time and the changes in the raw and logarithmic parameters were found (Table 5).

On the other hand, there were significant correlations between $T_{end}$ and the changes in the raw and the logarithmic parameters as estimated using the slope, except for LFB slopes estimated over longer periods than 50% of $T_{end}$ and DSI slopes over 100% $T_{end}$ (Table 6).

These results showed that the slopes in the logarithmic parameters correlated significantly with $T_{end}$. Table 6 shows that mean correlation coefficient of the raw parameters is 0.44 and that of the logarithmic ones 0.56, while the percentage increase was 26.3%. The predictor whose significant correlation was larger than 0.50 was the slope in logH/L-SM. Scatter plots of two predictors using the area ratio and the slope against $T_{end}$ and $\log(T_{end})$ are shown in Figures 11 and 12, comparing the two predictors using the raw and the logarithmic parameters. Mean correlation coefficients between $T_{end}$ and the slopes in the raw and the logarithmic parameters are shown in Figure 13 In a recent study, logH/L-SM was found to be more sensitive to muscle fatigue than the existing sEMG parameters such as RMS (root mean square), MPF and H/L-FB. It could be speculated that the more sensitive the parameter to muscle fatigue the better the correlation with endurance capacity.

**Table 5.** Correlation coefficients between the endurance times and area ratios as predictors estimated over the time periods of every 10% of $T_{end}$.

| (a) Raw parameter slopes v.s. $T_{end}$ | | | | | | | | | | | |
|---|---|---|---|---|---|---|---|---|---|---|---|
| **%$T_{end}$** | **10** | **20** | **30** | **40** | **50** | **60** | **70** | **80** | **90** | **100** | **Mean** |
| LFB | n.s. | n.s. | n.s. | n.s. | n.s. | n.s. | n.s. | n.s. | 0.44 [a] | 0.60 [b] | n.s. |
| MPF | n.s. | n.s. | n.s. | n.s. | n.s. | n.s. | n.s. | n.s. | n.s. | n.s. | n.s. |
| H/L-FB | n.s. | n.s. | n.s. | n.s. | n.s. | n.s. | n.s. | n.s. | n.s. | n.s. | n.s. |
| DSI | n.s. | n.s. | n.s. | n.s. | n.s. | n.s. | n.s. | n.s. | n.s. | n.s. | n.s. |
| H/L-SM | n.s. | n.s. | n.s. | n.s. | n.s. | n.s. | n.s. | n.s. | n.s. | n.s. | n.s. |
| Mean | n.s. | n.s. | n.s. | n.s. | n.s. | n.s. | n.s. | n.s. | n.s. | n.s. | n.s. |
| (b) Logarithmic parameter slopes v.s. log(Tend) | | | | | | | | | | | |
| **%$T_{end}$** | **10** | **20** | **30** | **40** | **50** | **60** | **70** | **80** | **90** | **100** | **Mean** |
| logLFB | n.s. | n.s. | n.s. | n.s. | n.s. | n.s. | n.s. | n.s. | n.s. | n.s. | n.s. |
| logMPF | n.s. | n.s. | n.s. | n.s. | n.s. | n.s. | n.s. | n.s. | n.s. | n.s. | n.s. |
| logH/L-FB | n.s. | n.s. | n.s. | n.s. | n.s. | n.s. | n.s. | n.s. | n.s. | n.s. | n.s. |
| logDSI | n.s. | n.s. | n.s. | n.s. | n.s. | n.s. | n.s. | n.s. | n.s. | n.s. | n.s. |
| logH/L-SM | n.s. | n.s. | n.s. | n.s. | n.s. | n.s. | n.s. | n.s. | n.s. | n.s. | n.s. |
| Mean | n.s. | n.s. | n.s. | n.s. | n.s. | n.s. | n.s. | n.s. | n.s. | n.s. | n.s. |

n.s.: non-significant; [a]: $p < 0.05$ indicates that the correlation coefficient is significant; [b]: $p < 0.05$ indicates that the correlation coefficient is significant and larger than 0.5.

**Table 6.** Correlation coefficients between the endurance times and slopes as predictors estimated over the time periods of every 10% of $T_{end}$.

| | | | | | | | | | | | |
|---|---|---|---|---|---|---|---|---|---|---|---|
| (a) Raw parameters slopes v.s. $T_{end}$ | | | | | | | | | | | |
| %$T_{end}$ | 10 | 20 | 30 | 40 | 50 | 60 | 70 | 80 | 90 | 100 | Mean |
| LFB | 0.42 [a] | 0.34 [a] | 0.36 [a] | 0.31 [a] | 0.31 [a] | n.s. | n.s. | n.s. | n.s. | n.s. | n.s. |
| MPF | 0.50 [b] | 0.43 [a] | 0.58 [b] | 0.56 [b] | 0.54 [b] | 0.60 [b] | 0.59 [b] | 0.60 [b] | 0.53 [b] | 0.50 [b] | 0.54 |
| H/L-FB | 0.42 [a] | 0.32 [a] | 0.44 [a] | 0.59 [b] | 0.45 [a] | 0.48 [a] | 0.38 [a] | 0.45 [a] | 0.48 [a] | 0.49 [a] | 0.45 |
| DSI | 0.43 [a] | 0.42 [a] | 0.42 [a] | 0.39 [a] | 0.41 [a] | 0.52 [b] | 0.53 [b] | 0.47 [a] | 0.32 [a] | n.s. | 0.41 |
| H/L-SM | 0.43 [a] | 0.45 [a] | 0.58 [b] | 0.54 [b] | 0.68 [b] | 0.69 [b] | 0.63 [b] | 0.64 [b] | 0.56 [b] | 0.47 [a] | 0.57 |
| Mean | 0.44 | 0.39 | 0.48 | 0.48 | 0.48 | 0.51 | 0.46 | 0.45 | 0.38 | n.s. | 0.44 |
| (b) Logarithmic parameters slopes v.s. log($T_{end}$) | | | | | | | | | | | |
| %$T_{end}$ | 10 | 20 | 30 | 40 | 50 | 60 | 70 | 80 | 90 | 100 | Mean |
| logLFB | 0.49 [a] | 0.48 [a] | 0.57 [b] | 0.69 [b] | 0.62 [b] | 0.65 [b] | 0.65 [b] | 0.64 [b] | 0.55 [b] | 0.47 [a] | 0.58 |
| logMPF | 0.57 [b] | 0.49 [a] | 0.62 [b] | 0.58 [b] | 0.57 [b] | 0.63 [b] | 0.57 [b] | 0.56 [b] | 0.45 [a] | 0.40 [a] | 0.54 |
| logH/L-FB | 0.52 [b] | 0.39 [a] | 0.47 [a] | 0.59 [b] | 0.59 [b] | 0.57 [b] | 0.41 [a] | 0.45 [a] | 0.45 [a] | 0.46 [a] | 0.49 |
| logDSI | 0.57 [b] | 0.53 [b] | 0.60 [b] | 0.60 [b] | 0.64 [b] | 0.66 [b] | 0.60 [b] | 0.58 [b] | 0.51 [b] | 0.45 [a] | 0.58 |
| logH/L-SM | 0.57 [b] | 0.55 [b] | 0.59 [b] | 0.60 [b] | 0.66 [b] | 0.70 [b] | 0.66 [b] | 0.65 [b] | 0.57 [b] | 0.51 [b] | 0.61 |
| Mean | 0.54 | 0.49 | 0.57 | 0.61 | 0.61 | 0.64 | 0.58 | 0.58 | 0.51 | 0.46 | 0.56 |

n.s.: non-significant; [a]: $p < 0.05$ indicates that the correlation coefficient is significant; [b]: $p < 0.05$ indicates that the correlation coefficient is significant and larger than 0.5.

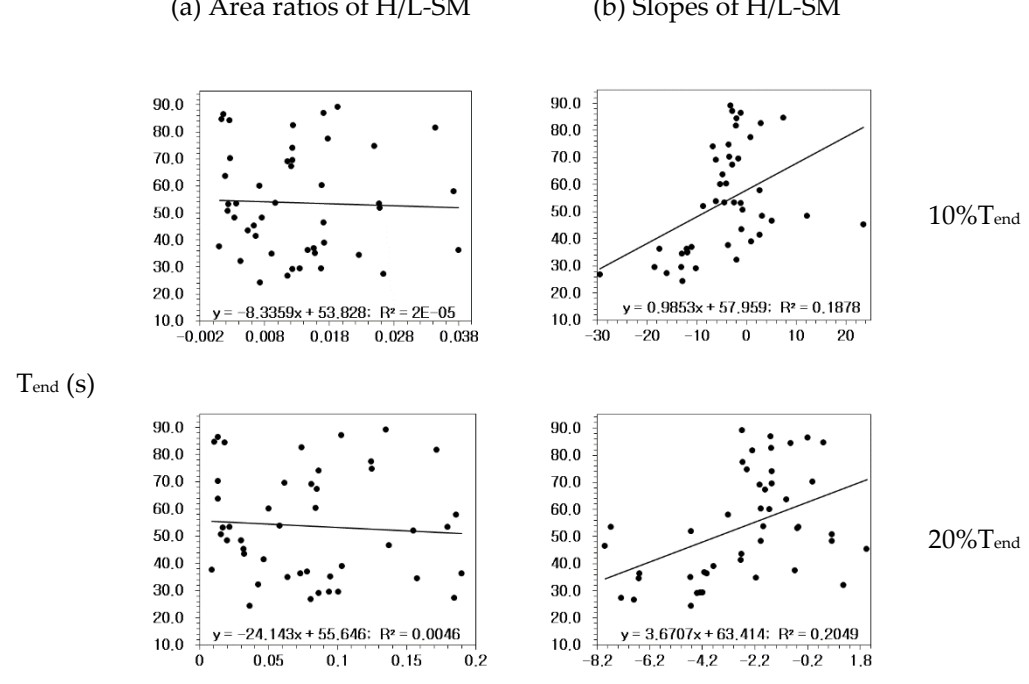

**Figure 11.** *Cont.*

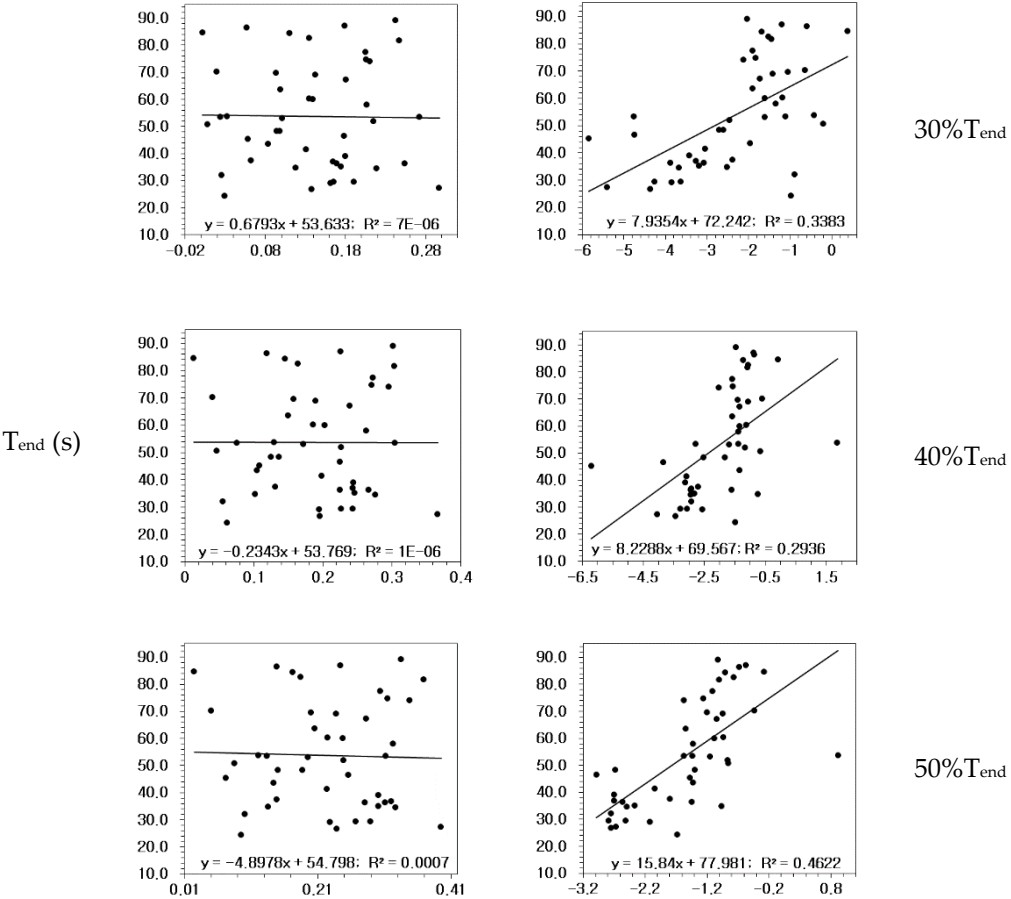

**Figure 11.** Scatter plots of the area ratio in the column (**a**) and the slope in the column (**b**) of the H/L-SM against $T_{end}$.

(a) Area ratios of log(H/L-SM))　　　(b) Slopes of log(H/L-SM)

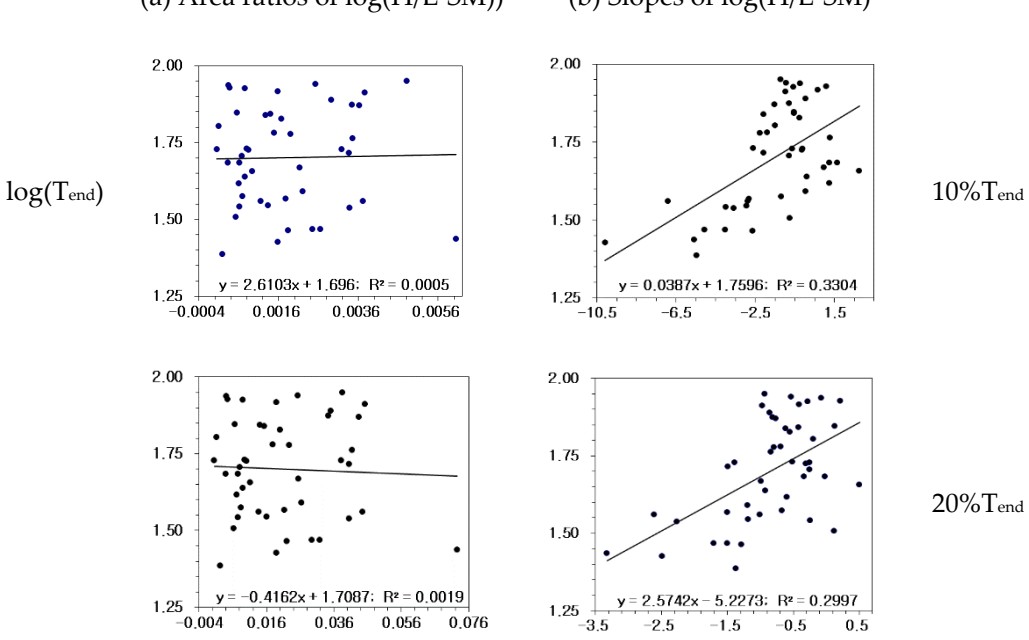

**Figure 12.** *Cont*.

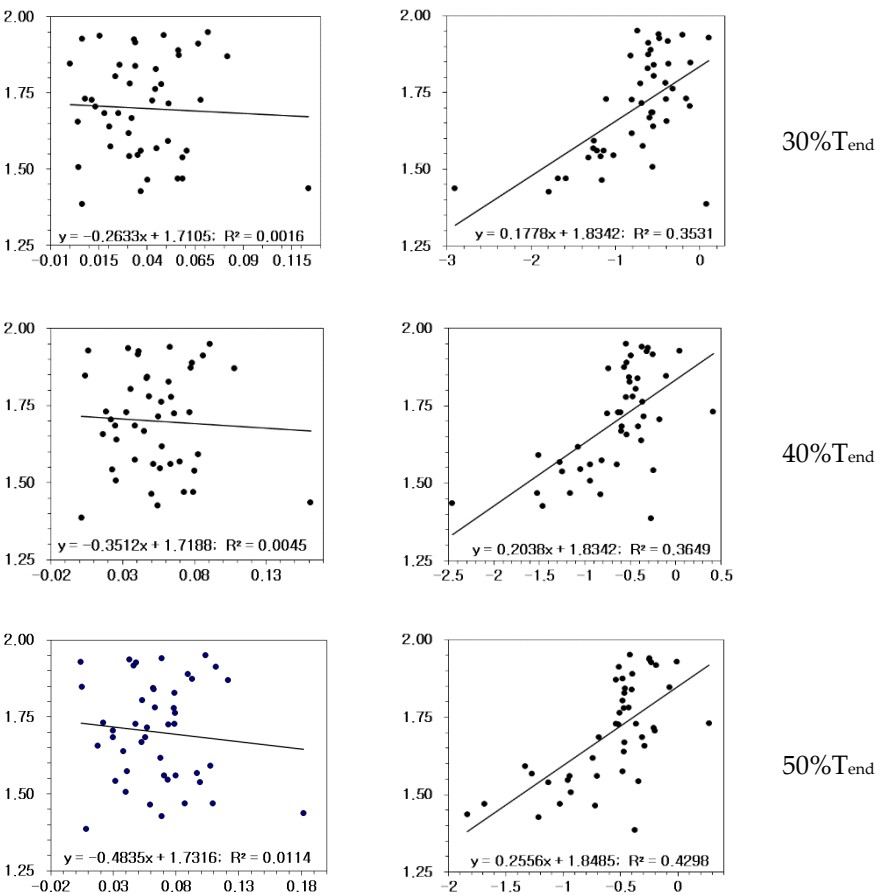

**Figure 12.** Scatter plots of the area ratio in the column (**a**) and the slope in the column (**b**) of logH/L-SM against log ($T_{end}$).

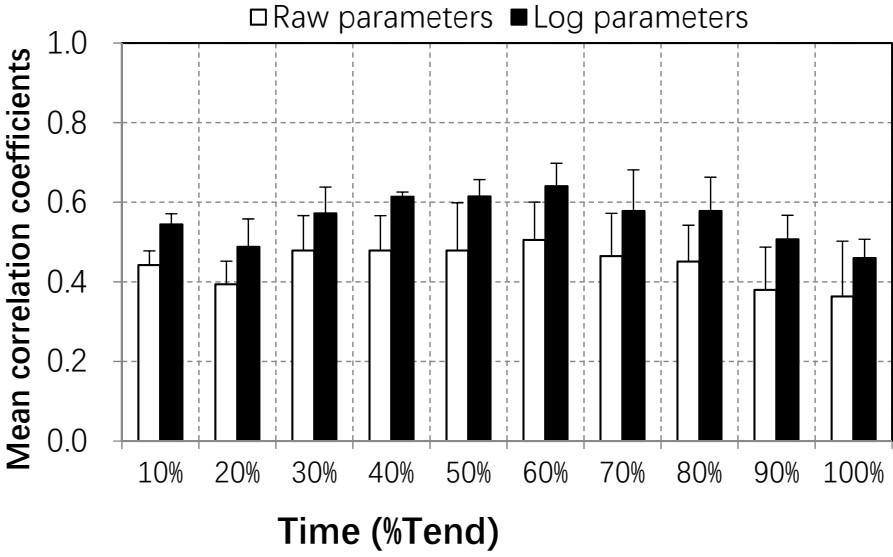

**Figure 13.** Comparison of mean and standard deviation of the correlation coefficients of raw and logarithmic parameters as described in Table 6 (a) and (b).

## 4. Discussion

The main object of this study was to assess the predictability of the endurance time using logarithmic parameters during isotonic contractions at low-moderate intensity. Previous studies had only focused on relatively high contraction levels which are generally uncommon in daily activities. In the present study, at 20% MVC the predictors such as the

area ratio and the slope using the raw and the logarithmic sEMG parameters were estimated from the onset to every 10% of $T_{end}$, and their relationships with $T_{end}$ were evaluated.

### 4.1. Changes in sEMG Parameters

The area ratio and the linear regressive slope as two statistical predictors of $T_{end}$ were used to estimate $T_{end}$ in this study. The area ratios varied linearly with respect to time (Figure 9). Maïsetti et al. (2002) obtained similar results showing that the area ratios in sEMG parameters were statistically linear with respect to time [8]. In contrast, the slopes varied exponentially with respect to time (Figure 10) in MPF and H/L-FB, while H/L-SM increased during the first part of the low-moderate level endurance test, probably because the behavior of the time series of the sEMG signal might be related to additional recruitment of motor units which occurred throughout sustained contractions at low contraction levels [40]. Similar results were obtained in the isometric endurance test of Van Dieën et al. (1998) and in the isotonic test of Lee et al. (2017) [20,23].

### 4.2. Comparison of Relationships

There were no significant correlations between $T_{end}$ and the predictor using the area ratio (Table 3), and scatter plots showed that the coefficients of determination were not enough to have significant correlation with $T_{end}$ (the column (a) in Figures 11 and 12). Maïsetti et al. (2002) also reported no correlations between $T_{end}$ and the area ratio in MPF, MDF, and MFCV with the exception of the highestt area ratio of LFB (6–30 Hz) observed for quadriceps muscles at 50% MVC [8]. In addition, Boyas et al. (2009) demonstrated that no correlations between $T_{end}$ and the changes in MPF using the area ratio were found, but there were significant correlations between $T_{end}$ and the changes in MPF using the linear regressive slope [38]. In the present study, when using the slope, we also found significant correlations between $T_{end}$ and the changes in the raw and the logarithmic parameters (Table 6), and the scatter plots showed their relationships (the column (b) in Figures 11 and 12), although spectral parameters have produced inconsistent trends during sustained contractions at low level.

As shown in Table 6, the mean correlation coefficient using the logarithmic parameters was increased by 26.3% compared to that using the raw parameters, and significant correlations larger than 0.50 were found between $\log(T_{end})$ and the slopes in logH/L-SM over the duration periods of every 10% of $T_{end}$. Previous studies showed that the slope in the spectral parameters as estimated over a shorter period than $T_{end}$ could be a suitable predictor and correlated with $T_{end}$ significantly. The logarithmic transformations were used to reduce the large variability in the raw parameters, and to show more sensitivity in localized muscle fatigue in sEMG-based assessments.

### 4.3. Limitations

The spectral parameters namely LFB, MPF, H/L-FB, DSI, and H/L-SM were extracted from the sEMG signal which, however, was non-stationary during the test, and wasaffected by many confounding factors [41], including electrode location, thickness of the subcutaneous tissues, the detection system used to obtain the recording, changes in the transmembrane action potential, and cross-talk from nearby muscles.

Due to all these confounding factors, caution is needed when using changes in the sEMG parameter as a predictor of the level of muscle fatigue. Although these factors could not be entirely excluded and do affect the estimated spectral parameters, we followed the recommendations that the careful placement of the electrodes between the innervation zone and the tendon and the normalized amplitudes should minimize the influence on the results.

The logarithmic transformation was used to convert the sEMG spectral parameters during the endurance dynamic contractions to minimize the effects of these factors [42]. MacIsaac et al. (2001) demonstrated that muscle fatigue could be assessed during dynamic contractions using a short-term Fourier transformation [43]. Some authors extracted

the signal spectral moments as spectral parameters using Fourier transformation [44]. Coorevits et al. (2008) found that continuous wavelet transformation and traditional Fourier transformation are generally reliable to assess muscle fatigue [45].

As mentioned in the introduction, the endurance capacity is problematic in measuring the effect of physical and psychological factors such as pain and motivation. Because many factors can influence sEMG spectral parameters, the correlation of their changes with endurance might be lower than expected.

## 5. Conclusions

The present study demonstrated that the slope of the logarithmic parameter was a suitable predictor for monitoring biceps brachii muscle fatigue and for predicting $T_{end}$, even when it was estimated over every 10% of $T_{end}$ during isotonic fatiguing contractions at a low-moderate level. The main conclusions of this study can be stated as follows:

(1) The linear regressive slope was a more suitable predictor of $T_{end}$ than the area ratio.
(2) Significant correlations using the logarithmic parameters were about 26.3% higher than those using the raw ones.
(3) Significant correlations larger than 0.5 were found between $\log(T_{end})$ and the slopes of logH/L-SM over a duration time of every 10% of $T_{end}$.

From a clinical perspective, this sEMG method is useful to predict $T_{end}$ compared to the mechanical method of measuring $T_{end}$, in reducing the length of the endurance test and in minimizing the influence of physical and psychological factors. Further studies are needed to evaluate this method for the muscles of the lower limbs and to develop the predictability using a combination of the parameters.

**Author Contributions:** Conceptualization, S.-S.L.; Formal analysis, J.-H.J.; Methodology, J.H.J.; Resources, Y.-j.K.; Visualization, C.-o.C.; Writing—review & editing, K.-y.L. All authors have read and agreed to the published version of the manuscript.

**Funding:** This study was supported by the Catholic Kwandong University (202001900001).

**Institutional Review Board Statement:** Approved by the Ethics Committee.

**Informed Consent Statement:** Informed consent was obtained from all subjects involved in the study.

**Data Availability Statement:** MDPI Research Data Policies.

**Conflicts of Interest:** The authors declare no conflict of interest.

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
