# Peer review of "Comparison of Endurance Time Prediction of Biceps Brachii Using Logarithmic Parameters of a Surface Electromyogram during Low-Moderate Level Isotonic Contractions"

_applsci, doi:10.3390/app11062861_

Round 1

Reviewer 1 Report

Overview and general recommendation:

Surface electromyography is applied as an assessment tool in the prevention, monitoring and rehabilitation fields. Because of this, there is a need of defining normative indicators or parameters during activities that requires low muscle activity. So that, there is a research gap and because of this, the current perspective is on a topic of relevance and general interest to the readers of the journal.

However, although the topic of the paper is very interesting, I found some concerns about its acceptance in its current form.  I detail my concerns below.

Major comments:

  1. In general, the paper is difficult to follow and understand. An extensive editing of English language and style is required
  2. Page 3. sEMG electrodes were placed in the biceps muscle. However, where was its exact location? Did you follow any recommendations? Please, add the references followed for its placement.
  3. Page 6. You say “The logarithmic transformation have been widely used in biomedical and psychosocial re-search to deal with inconsistent data”. Please, add references that confirm this hypothesis.
  4. Page 7. Data analysis. You segmented data using 1 second and 0.3 seconds. Why did you choose these values? Did you rely on other works? If yes, please add references. If no, add the reasons of using these values.
  5. Page 7. Data analysis. What is the coefficient of variation? What is its mathematical definition?
  6. In general, the captions of the figures are very short and figures are low-quality images. All figures must be legible, self-contained and look good.
  7. Figure 7. the x-axis is named subjects but you only had 10 subjects and the figure shows until 50. What is wrong?
  8. Table 2 and figures 9, 10 and 13 show mean values. Could you also show the standard deviation of these values?
  9. Some tables and figure axes missed units. Please add them

Reviewer 2 Report

The work presented by Chang-ok Cho and colleagues showed a comparison of endurance time prediction of biceps brachii using logarithmic parameters of surface electromyogram during low-moderate level isotonic contractions. This study is novel, and the manuscript is well written. However, I just have some minor concerns.

Several references are missing, and some are not formatted correctly, for example-

“In everyday life, low-moderate level isotonic exercise is the natural way of human activity which includes a concentric contraction and an eccentric contraction. Concentric contractions are the primary functions of biceps brachii muscles, and endurance contractions primarily work the slow twitch fibers and develop such fibers in their efficiency and resistance to fatigue.”

“Badier et al (1993) found a significant relationship between Tend and the time-constant of high-to-low ratio with fixed frequency band as computed within the first 10-20 s of contraction. Hanayama (1994) found that no significant correlation beween Tend and the decreasing changes of muscle fiber conduction velocity (MFCV).”

In the Materials and Methods Section, please write how the sample size was calculated.

Round 2

Reviewer 1 Report

The authors responded to the majority of my comments.